# Productive Performances of Slow-Growing Chicken Breeds and Their Crosses with a Commercial Strain in Conventional and Free-Range Farming Systems

**DOI:** 10.3390/ani13152540

**Published:** 2023-08-07

**Authors:** Edoardo Fiorilla, Marco Birolo, Ugo Ala, Gerolamo Xiccato, Angela Trocino, Achille Schiavone, Cecilia Mugnai

**Affiliations:** 1Department of Veterinary Sciences, University of Turin, Largo Braccini 2, 10095 Grugliasco, Italy; ugo.ala@unito.it (U.A.); achille.schiavone@unito.it (A.S.); cecilia.mugnai@unito.it (C.M.); 2Department of Agronomy, Food, Natural Resources, Animals and Environment, University of Padova, Viale dell’Università, 16, 35020 Padova, Italy; marco.birolo@unipd.it (M.B.); gerolamo.xiccato@unipd.it (G.X.); angela.trocino@unipd.it (A.T.); 3Department of Comparative Biomedicine and Food Science, University of Padova, Viale dell’Università, 16, 35020 Padova, Italy

**Keywords:** local chicken breeds, crossbreeding, free-range, sustainability, low input diets

## Abstract

**Simple Summary:**

Poultry farming is set to expand in the future, and consumers are requiring more sustainable and ethical products. Free-range farming systems produce high-quality meat and also present added value in terms of welfare, sustainability, small-scale farmers’ development, and ethical farming. Local chicken breeds should be used in organic and free-range farming, but they have long growth periods. As found in the present study, crossbreeding could improve growth results, thus safeguarding local chicken biodiversity.

**Abstract:**

Local chicken breeds play a vital role in promoting sustainability by preserving genetic diversity, enhancing resilience, and supporting local economies. These breeds are adapted to local climates and conditions, requiring fewer external resources and inputs for their maintenance. By conserving and utilizing local chicken breeds, sustainable farming practices can be incentivized, maintaining ecosystem balance and ensuring food security for future generations. The present study aimed at evaluating the growth performance and slaughter traits of two local Italian chicken breeds (Bionda Piemontese and Robusta Maculata) and their crosses with a medium-growth genotype (Sasso chicken^®^) reared in conventional and free-range farming systems. The conventional system used a high-energy high-protein diet in a closed barn with controlled temperature, humidity, and lighting, and a stocking density of 33 kg/m^2^. The free-range system used a low-input diet (low-energy low-protein diet composed of local and GMO-free feed ingredients), uncontrolled environmental conditions, and a stocking density of 21 kg/m^2^ in a barn with free access to an outdoor area. The birds were slaughtered at 84 days of age in both systems. The crossbred chickens showed the best results for growth performance in both farming systems compared to local breeds. Within genotype, the final live weight of chickens was similar in the two farming systems. In conclusion, slow-growth crossbreeds should be used in alternative farming systems, demonstrating better performance than pure local breeds.

## 1. Introduction

Poultry farming is considered to be of great importance for human nutrition, especially in the framework of developing countries and the economic crisis linked to overpopulation [1]. Among livestock species, poultry has the smallest environmental impact, with an output of 0.1 gigatons of carbon dioxide compared to the 1.8 gigatons emitted by cattle breeding [2]. Furthermore, poultry meat has a high nutritional value, is free from religious restrictions, and represents a relatively cheap protein source compared to other livestock [3]. This outlook is confirmed by the data provided by OECD-FAO reporting that every year, world meat production undergoes a contraction for almost all the traditionally farmed species except poultry, which is constantly growing, even during the recent COVID-19 pandemic [4,5].

Conventional farming systems, commonly called intensive, are designed to maximize efficiency and yield [6]. In these systems, chickens are commonly housed in large numbers in closed barns often equipped with artificial lighting and controlled temperature and ventilation [7]. The animals are usually fed a high-energy diet in order to promote rapid growth [8]. Intensive breeding systems can provide high yields and could be considered economically convenient, even though these results are obtained at the expense of animal welfare, as crowded living conditions and a lack of natural behaviors can lead to stress and health problems [9].

Free-range farming systems, on the other hand, could be complementary to conventional rearing [10]. In this system, animals have access to an outdoor area where they can roam and forage. This allows the chickens to express natural behaviors, providing them with a better quality of life [11]. However, free-range systems require more land and often require more labor than conventional methods [12]. Overall, free-range systems prioritize animal welfare and sustainability over maximum yields and efficiency, and although this may be considered economically disadvantageous for some consumers, it could represent an added value as it would be a more acceptable and sustainable approach to poultry production [13].

Driven by consumer demand for healthy and environmentally friendly food, organic farming in Europe has gained momentum in recent years [14]. The European Union (EU) has set very strict regulations and standards for organic farming. To be certified organic, farms must adhere to these regulations, which include a minimum conversion period from conventional to organic farming, regular inspections, and documentation of their farming practices (Regulation (EU) No. 2018/848 of 30 May 2018).

Free-range farming systems, on the other hand, are not as regulated, since only the Commission Regulation (EC) No. 543/2008 of 16 June 2008 dictates some indications and rules, and a complete and comprehensive set of rules for “free-range” livestock production does not exist yet [15].

In this context, the use of the most suitable chicken breed is essential to optimize not only the productivity but also the welfare of the poultry in each farming system [10]. Different breeds of chickens have been selectively bred over the years for specific traits such as egg or meat production, resistance to disease, and adaptability to specific environments. Therefore, selecting the right breed for a particular husbandry system could help ensure that birds are suited to the conditions in which they are raised and can thrive while optimizing production [10,16]. Commercial hybrid breeds are usually bred for high productivity in intensive farming systems but are not ideal for free-range systems [17]. In free-range systems, breeds that are more active, adaptable, and resistant to environmental stress are needed to thrive. Therefore, local breeds may be better suited for free-range systems, as they are typically more adaptable to outdoor environments and have better natural instincts for foraging and predator avoidance [18,19].

Crossbreeding between chickens is a common practice used by farmers to improve the production performance and resilience of their flocks [20]. By selectively breeding different breeds of chickens, farmers can create offspring that possess desirable traits, such as higher egg production and faster growth rates, combined with good resistance and adaptability to stressors. Crossbreeding also allows farmers to introduce genetic diversity into their flocks, which could help reduce the risk of inbreeding [21]. However, this useful tool must be used in the correct way by organizing proper mating programs trying to protect the pure local breed as well.

In the overall vision of a free-range farming system, it is also necessary to consider the diet supplied to the animals. Low-input diets in poultry farming may represent a sustainable approach to poultry production to minimize the use of costly and resource-intensive inputs such as commercial feed, which is rich in soy and high-energy cereals [10]. These low-input diets favor the use of locally available ingredients like legumes and vegetable byproducts, which could help reduce the environmental impact of poultry production [22]. Furthermore, in a free-range farming system, given the breeds used, it is not necessary to provide an extremely energetic and rich-in-protein diet, as it would be a waste to supply it to animals not able to take full advantage of this energy surplus [23].

Thus, safeguarding local poultry breeds could guarantee sustainable, healthy, and noble proteins in long-term human nutrition. We are still doing too little to mitigate the loss of biodiversity, and the fragility of ecological systems is visible on a global scale. The extraordinary genetic variability and adaptability of local breeds must be preserved. Bionda Piemontese is a breed protected by a consortium of breeders who guard the social and culinary traditions of their own breeds, and Robusta Maculata is protected by the Veneto Regional Agriculture Association. Even though we have some data on these local breeds, there are no studies aiming at improving the growth performance of these animals with crossbreeding programs.

As crossbreeding is expected to improve growth performance, the aim of this trial was to study the response of two local Italian chicken breeds and their crossbreeds to commercial and low-input diets and different farming systems in order to explore the possibility to sustain and valorize the local breeds.

## 2. Materials and Methods

The experimental protocol was approved by the Bioethical Committee of the University of Turin (Italy) (Prot. ID: 251833). All animals were reared, managed, and processed according to regulation 2007/43/EC for the protection of chickens kept for meat production and regulation 2010/63/EU for the protection of animals used for scientific purposes.

The trial was performed in the poultry facility of the Department of Veterinary Sciences of the University of Turin (Italy) in the spring of 2021.

### 2.1. Animals, Facilities, and Experimental Design

Four different poultry genotypes were used: two local Italian pure breeds, namely, Bionda Piemontese (BP) and Robusta Maculata (RM), and their F1 crossbreed with the Sasso (medium growing hybrid genotype, Hendrix genetics^®^), resulting in Bionda Piemontese × Sasso (BP × S) and Robusta Maculata × Sasso (RM × S).

Bionda Piemontese is a breed native to the Piedmont region (Northwest Italy), characterized by golden plumage and a black tail. It is considered a dual-purpose breed as females can lay up to 180–200 eggs a year and males are used for meat production. The preferred slaughtering age is at around 24 weeks of age [24]. Robusta Maculata is a breed native to the Veneto region (Northeast Italy), characterized by silvery white plumage with fringes and irregular black spots. It is also considered a dual-purpose breed: females can lay up to 140–160 eggs a year while males are used for meat production; they are slaughtered at around 22 weeks of age and can weigh up to 3.8–4.4 kg [25]. The breeders of BP and RM are included in the consortium for the conservation of the breed and in the biodiversity conservation project of Italian poultry breeds sponsored by the Ministry of Policies Agricultural, Food, and Forestry [26].

The Sasso chicken is a breed of meat chicken that was developed in France in the 1960s (Hendrix Genetics^®^). It is a genetic hybrid created by crossing different strains of chickens to obtain animals that grow quickly and have a good meat yield. Sasso chickens are known for their good size, tender meat, and medium growth rate, which make them a popular choice for organic meat production. They have a calm temperament and are adaptable to a variety of environments, which also makes them a good choice for backyard chicken keepers.

The eggs of BP were provided by the Avian Conservation Centre for Local Genetic Resources of the University of Turin (Italy) located in Carmagnola (Turin, Italy), while the eggs of RM were provided by Veneto Agricoltura (Padova, Italy). The eggs of the two crossbreed genotypes were provided by the project partner, the University of Perugia (Italy), using roosters of local breeds and hens of Sasso (Ruby C strains). All the eggs were moved and incubated at the same time in a local commercial incubator (Monge, Torre San Giorgio, Italy).

One-day-old chicks were moved to the experimental poultry facility and reared from hatching until reaching 20 days of age in the brood, which was divided into four pens (one for each genotype). The pens were 1 m wide and 2 m long, with net walls and a waterproof floor covered with wood shavings as litter (20 cm high). The brood was environmentally controlled, with temperature and relative humidity (RH) ranging from 32 to 20 °C and from 70 to 65%, respectively. The lighting schedule was 23 h light/1 h dark until day 3, and then the dark period was gradually increased to 6 h. The environmental parameters were monitored daily during the whole period of the trial.

At 21 days of age, 264 chicks were selected based on mean weight within the genotype (BP: 233 g; RM: 207 g; BP × S: 271 g; RM × S: 258 g), individually labeled with a wing mark, and allotted to the two different farming systems (conventional and free-range) with 3 replicate pens per treatment for a total of 24 pens (Table 1).

Due to the different expected weights at slaughter between the purebred and crossbred genotypes and to maintain a similar stocking density at slaughter weight (33 kg/m^2^ in the conventional system and 21 kg/m^2^ in the free-range system), 18 and 15 chicks/pen for purebred and crossbred genotypes, respectively, were allocated to the conventional system, and 6 and 5 chicks/pen for purebred and crossbred genotypes, respectively, were allocated to the free-range system.

In the conventional farming system (C), the lighting schedule was 18 h light/6 h dark for the whole trial. Temperature and relative humidity in the poultry house were set according to Aviagen guidelines [27]. A total of 12 pens were prepared with three replicates for each of the four genotypes (Table 1), with a final stocking density of 33 kg/m^2^. Each pen was equipped with wood shavings as litter.

In the free-range farming system (F), the birds were exposed to a natural temperature and photoperiod. The mean temperature in the poultry house during daylight hours was 21 °C, and at night it was 15 °C. According to the season, the mean hours of daylight during the trial were 14 h/day. The birds were divided into 12 pens with three replicates for each of the four genotypes (Table 1). The poultry house was divided into an indoor and an outdoor area: the indoor pens were equipped with wood shavings as litter and the final stocking density was set at 21 kg/m^2^; outdoor the animals had 10 m^2^ available surface per animal according to European Council (EC) Regulation No. 543/2008. The animals were free to stay either outside or inside at any time of the day. The whole facility, including the outdoor areas, was protected from wild birds and predators with fences. Water was administered ad libitum and a specifically formulated low-input diet with reduced soybean meal in favor of local ingredients, like faba bean and GMO-free organic soybean meal, was provided (Table 2).

Mortality and health status were checked daily during the whole experimental period. All the birds were individually weighed. Slaughtering was performed according to the EU recommendation for organic poultry production, i.e., at 12 weeks (84 days old) [28].

### 2.2. Diets

Three different experimental feeds were provided to the birds: a common starter diet for the first 20 days in the brood, a standard diet for the conventional farming system, and a low-input diet for the free-range farming system from 21 days of age onwards (Table 2).

The diets were the same as in a previous study, and the complete formulation can be found in the work by Fiorilla et al. [29].

The starter and the standard diets were formulated to meet the energy and protein levels commonly recommended for conventional farming systems [27]. The low-input diet was administered to the birds reared in the free-range system and was formulated with the aim of reducing the protein content and shortening the use of imported soybean meal and replacing it with faba bean and GMO-free soybean meal coming from local Italian cultivations. All the diets were pelleted and produced by an industrial feed mill (Cortal Extrasoy S.P.A., Cittadella, Padova, Italy).

The experimental diets were ground to pass through a 0.5 mm sieve and stored in airtight plastic containers. The experimental diets were analyzed for DM (method number #934.01), ash (method number #942.05), CP (method number #984.13), and EE (method number #2003.05) according to AOAC International [30]. All analyses were performed in triplicate.

**Table 2 animals-13-02540-t002:** Diet ingredients and composition of each diet and farming system.

	Starter Diet (1–20 d)	Grower Diets (21–84 d)	
		Standard	Low-Input
Chemical Composition			
Dry matter (%)	89.13	88.34	88.78
Crude protein (%)	21.26	19.49	16.59
Ether extract (%)	6.14	7.36	5.23
Crude fiber (%)	3.17	1.72	2.68
Ash (%)	6.15	6.02	6.06
Lysine ^§^ (%)	1.20	1.07	0.95
Methionine + cysteine ^§^ (%)	0.71	0.66	0.53
Calcium ^§^ (%)	1.01	1.00	0.99
Phosphorus ^§^ (%)	0.70	0.68	0.68
Metabolizable energy (kcal/kg) ^§^	3089	3252	2921

Feed composition and nutritional additives by Fiorilla et al. [29]. ^§^ Estimated values.

### 2.3. Growth Performances

Starting from 21 days of age, all the chickens were weighed weekly. The feed was removed 2 h before the birds were weighed. The individual live weight (LW) was recorded using an electronic scale (KERN PLE-N v. 2.2) by gently placing the chicken inside a dark container and then onto the scale. Feed conversion ratio (FCR), average daily weight gain (ADG), and daily feed intake (ADFI) were calculated for each week and for the overall experimental period on a pen basis. Mortality and clinical signs of illness were monitored daily throughout the trial.

### 2.4. Slaughtering and Carcass Dissection

At 84 days of age, three birds per pen (i.e., 72 birds, 9 per experimental group) were selected as representatives of the average live weight and standard deviation of their pen and slaughtered after 12 h of feed withdrawal in a commercial abattoir. Live weight was recorded before the birds were electrically stunned and slaughtered. After death, carcasses were plucked, eviscerated (non-edible viscera: intestines, proventriculus, gall bladder, spleen, esophagus, and full crop), and stored for 24 h at +4 °C. Head, neck, legs, edible viscera (heart, liver, gizzard), and fat (perivisceral, and abdominal) were removed in order to obtain the ready-to-cook carcass (RTCC) [31]. The weight of the heart, spleen, liver, and gizzard were recorded and the data were expressed as a percentage of LW. The gizzard was emptied and then weighed to obtain the net weight of the organ. The chilled carcass (CC) weight was registered after storage at +4 °C for 24 h. The breasts and thighs were then excised, and their weights were expressed as a percentage of the CC weight.

### 2.5. Statistical Analysis

For the growth performance analysis, each pen served as an experimental unit with three pens per treatment. The slaughtering performance analysis utilized individual birds as the experimental unit. Levene’s test was employed to establish the homogeneity of variance, while the Shapiro–Wilk test determined the normality or non-normality of distribution. The analysis of growth performance, slaughtering yield, and carcass quality was conducted using a generalized linear mixed model (GLMM). The GLM considered three fixed factors: genotype, farming system, and the interaction between genotype and farming system. The replicate was included as a random effect to account for repeated measurements on the same pen. The “Identify_outliers” package was used to highlight and remove possible outliers, the Shapiro–Wilk test to assess the normality of the distribution of data, and Levene’s test to check for equality of variances. *p*-values are considered significant when *p* < 0.05. Data were reported as mean values with SEM. Analyses were conducted in R (version 3.6.3) and R Studio (Version 1.2.1335).

## 3. Results

During the whole experimental period, the animals showed no signs of illness or sickness, and only one bird of the Robusta Maculata breed died in the conventional farming system at 61 days of age.

### 3.1. Growth Performance

The data related to the weekly growth performance traits are shown in Figure 1 and Figure 2. The two crossbreeds started showing higher weights around 42 days of age (Figure 1 and Figure 2). In the F system, RM had the lowest weights at 63, 70, and 77 days of age (Figure 2), while no differences between BP and RM were found in the C farming system.

The final growth performance traits can be found in Table 3. Overall, no differences were highlighted in LW and ADG between the two farming systems (*p* > 0.05), whereas LW and ADG were significantly lower in the two purebreds (BP, RM) compared to the two crossbreeds (BP × S, RM × S) (*p* < 0.05), without significant effects of the genotype × farming system interaction. No significant interaction was found for both farming systems (*p* > 0.05).

ADFI did not differ between genotypes (*p* > 0.05), while higher feed consumption was recorded in chickens kept in the free-range system compared to the conventional system. Moreover, a significant interaction between genotype and farming system was recorded, with higher results in the free-range farming system (*p* < 0.05). Finally, as for FCR, it was significantly higher in local breeds compared to crossbreeds and in chickens kept in the free-range system compared to those in the conventional one (*p* < 0.05).

### 3.2. Slaughtering Performances

The data related to slaughtering performances can be found in Table 4.

Breast yields followed the pathway of LW, with higher values for BP × S and RM × S chickens compared to BP and RM chickens (*p* < 0.05). No effect of the farming system and its interaction with genotype was detected (*p* > 0.05). Thigh yields were similar among genotypes but with significantly higher values for all four experimental groups in the free-range farming system (*p* < 0.05).

Spleen, heart, liver, gut, and gizzard showed no differences between genotypes or farming systems (*p* > 0.05). The gizzard was the only one which had similar results between genotypes but with significantly higher values for all four experimental groups in the free-range farming system (*p* < 0.05).

## 4. Discussion

The key role of poultry farming in feeding the world is widely recognized and it is therefore necessary that it evolves along with society by listening to the requests of consumers who are increasingly interested in animal welfare, adapting to the need to mitigate the environmental impact of animal husbandry and help the development of different farming sectors that can stimulate and improve each other [32,33,34].

In the present trial, local breeds reached weights comparable to those found in previous studies on the same breeds [24] and other breeds from different geographical areas [35]. They are clearly not suited for conventional farming and will never be able to perform like commercial hybrids, but their characteristics of resilience, good meat quality, and ethical and cultural value are inestimable. Local breeds represent an invaluable heritage in biodiversity and their use in crossbreeding could allow us to preserve the nature and purity of local breeds, increasing their population with a subsequent decrease in consanguinity and related problems [36]. The results of this study confirmed that crossbreeding can significantly improve the growth performance of chickens, as largely known [37]. Under our conditions, the crossbred chickens showed better growth rates, higher breast meat yield, and better feed efficiency than purebred chickens while maintaining adaptability, as no mortality occurred in the free-range breeding system.

Crossbreeds presented a regular weight increase during the trial, unlike the local breeds. In fact, BP had been found to show a fast growth rate within the first 7–8 weeks that slows down and reaches a plateau in the following period [24,29]. The RM, being morphologically a heavier and larger breed than the BP, seems to need more time to build up muscle. In fact, we found BP to be in the lead in terms of weight for the first 8–9 weeks to match later RM between weeks 10 and 12. A careful analysis of the chosen breeds and the knowledge of them can give indications of the most useful characteristics to improve. For example, BP chickens seem to be more suitable for shorter periods of production; therefore, the aim could be to improve their performance within the first 90 days of life. RM chickens, on the other hand, seem to take longer periods to grow but reach higher LW than BP, so it would be necessary to aim for an improvement in the growth rate within the first 110 days, balancing their slower growth rate in the first weeks of age.

However, the results obtained from BP × S and RM × S chickens show how these animals, without being completely changed, can fit into a niche in the poultry industry. In fact, the aim is not to replace conventional breeding—which, as it is known, allows a large part of the population to be fed in a very short time at low energy and environmental costs—but to augment it by offering quality products obtained considering animal welfare and giving the product an ethical value that many consumers have recently begun to require.

As for the farming systems, the growth results obtained in the free-range system are comparable to the conventional system despite no environmental controls and the animals being fed a diet with a lower protein and energy content than those in the conventional system. However, these growth benefits must be managed and exploited correctly, taking into consideration the safeguarding of local breeds and their genetic heritage. It is therefore essential to identify the right chicken for the right farming system, as previous research already demonstrated [10,38,39].

The analysis of the carcass yields and main cuts showed that birds were able to maintain comparable data between farming systems, increasing only the thigh yields and gizzard size in the case of the free-range system. The first results could be justified by the greater kinetic activities of free-range birds, thanks to the higher space allowance, which likely promoted thigh muscle development [40,41,42]. As for the increase in gizzard size of chickens kept in the free-range system, the presence of grass in the external paddock combined with the possibility of scratching likely accounted for this result [43,44].

## 5. Conclusions

The present study verified the possibility of improving local chicken breeding in terms of F1 outputs by crossing local breeds with medium-growth genotypes. Based on these results, F1 outputs showed to be well adapted to the free-range system compared to the conventional system. In fact, in the conventional system, with higher energy and protein levels but poor space availability for kinetic activities, these birds did not increase productive performance.

The next step necessary to confirm this adaptability is a future study on the well-being and behavior of these animals in order to confirm whether a free-range farming system with environmental enrichment can actually benefit the animals’ well-being.

Future in-depth studies on crossbreeding could help maximize the performance of these local breeds. This would not only make it possible to increase the choice for farmers, who could decide to breed the animals that are best suited to their geographical and climatic area, but it would also stimulate greater competition with the large industrial players. Furthermore, it could also help the farmers who have been involved in saving the local breeds by providing them with the funds to increase their efforts and improve their facilities.

## Figures and Tables

**Figure 1 animals-13-02540-f001:**
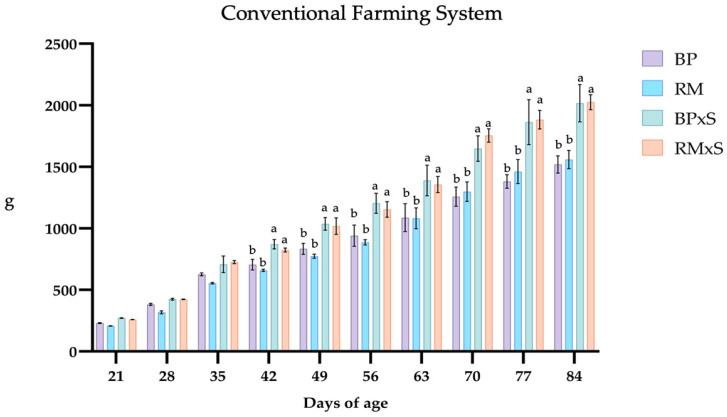
Weekly recorded live weight (g) of the four genotypes reared in the conventional system. Abbreviations: BP, Bionda Piemontese; RM, Robusta Maculata; BP × S, Bionda Piemontese × Sasso; RM × S, Robusta Maculata × Sasso. a, b values differ significantly. *p*-values are considered significant when *p* < 0.05.

**Figure 2 animals-13-02540-f002:**
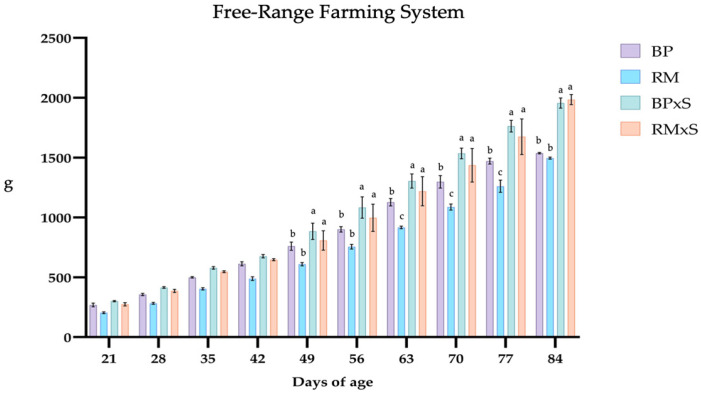
Weekly recorded live weight (g) of the four genotypes in the free-range system. Abbreviations: BP, Bionda Piemontese; RM, Robusta Maculata; BP × S, Bionda Piemontese × Sasso; RM × S, Robusta Maculata × Sasso. a, b, c values differ significantly. *p*-values are considered significant when *p* < 0.05.

**Table 1 animals-13-02540-t001:** Experimental design of trial and animal distribution in the farming systems.

Genotype	Conventional (C)	Free-Range (F)
Bionda Piemontese	3 pens with 18 chicks/pen (33 kg/m^2^)	3 pens with 6 chicks/pen (21 kg/m^2^)
Robusta Maculata	3 pens with 18 chicks/pen (33 kg/m^2^)	3 pens with 6 chicks/pen (21 kg/m^2^)
Bionda Piemontese × Sasso	3 pens with 15 chicks/pen (33 kg/m^2^)	3 pens with 5 chicks/pen (21 kg/m^2^)
Robusta Maculata × Sasso	3 pens with 15 chicks/pen (33 kg/m^2^)	3 pens with 5 chicks/pen (21 kg/m^2^)

**Table 3 animals-13-02540-t003:** LW (g), ADFI (g/d), ADG (g/d), and FCR of the four genotypes in the two farming systems.

	Conventional System				Free-Range System				SEM			*p*-Value		
	BP	RM	BP × S	RM × S	BP	RM	BP × S	RM × S	G	FS	G × FS	G	FS	G × FS
LW 21 d (g)	233	207	271	258	234	205	275	256	1.98	2.04	2.11	0.679	0.912	0.881
LW 84 d (g)	1519 ^b^	1558 ^b^	2016 ^a^	2025 ^a^	1537 ^b^	1495 ^b^	1955 ^a^	1984 ^a^	11.53	12.21	12.96	0.043	0.351	0.565
ADFI g/d	72.7 ^b^	74.1 ^b^	76.3 ^b^	79.4 ^b^	95.7 ^a^	97.7 ^a^	94.6 ^a^	93.2 ^a^	1.6	1.6	1.9	0.729	0.036	0.028
ADG g/d	19.8 ^b^	20.8 ^b^	25.6 ^a^	25.4 ^a^	20.3 ^b^	19.3 ^b^	25.1 ^a^	24.6 ^a^	0.33	0.37	0.38	0.046	0.493	0.515
FCR	4.77 ^b^	4.62 ^b^	3.86 ^a^	3.67 ^a^	5.53 ^d^	5.67 ^d^	4.65 ^c^	4.51 ^c^	0.13	0.16	0.24	0.048	0.036	0.021

Abbreviations: SEM, standard error of the mean; G, genotype; FS, farming system; G × FS, interaction between genotype and farming system; LW, live weight; ADFI, average daily feed intake; ADG, average daily weight gain; FCR, feed conversion ratio; BP, Bionda Piemontese; RM, Robusta Maculata; BP × S, Bionda Piemontese × Sasso; RM × S, Robusta Maculata × Sasso. Values with superscript ^a^, ^b^, ^c^, ^d^ are considered significantly different. *p*-values are considered significant when *p* < 0.05.

**Table 4 animals-13-02540-t004:** Effect of genotype and farming system on slaughtering traits of birds.

	Conventional System				Free-Range System				SEM			*p*-Value		
	BP	RM	BP × S	RM × S	BP	RM	BP × S	RM × S	G	FS	G × FS	G	FS	G × FS
SW (g)	1576	1592	2006	2038	1528	1512	2011	1993	2.02	1.99	2.08	0.032	0.534	0.349
RTCC (%SW)	65.59	65.15	66.21	66.15	65.32	65.23	66.17	66.19	0.25	0.31	0.31	0.516	0.628	0.552
CC (% SW)	63.63	62.24	62.39	62.21	63.20	62.69	65.19	64.33	0.78	0.92	1.18	0.219	0.362	0.505
Breast (% CC)	11.64 ^b^	11.16 ^b^	13.85 ^a^	13.56 ^a^	10.11 ^b^	10.02 ^b^	13.82 ^a^	13.54 ^a^	0.34	0.32	0.46	0.043	0.139	0.346
Thigh (% CC)	27.72 ^b^	27.55 ^b^	28.46 ^b^	28.90 ^b^	33.45 ^a^	33.98 ^a^	34.16 ^a^	35.22 ^a^	0.28	0.24	0.24	0.218	0.022	0.019
Spleen (% SW)	0.15	0.16	0.17	0.17	0.16	0.17	0.19	0.19	0.01	0.01	0.01	0.432	0.328	0.418
Heart (% SW)	0.51	0.51	0.53	0.54	0.52	0.52	0.54	0.54	0.02	0.02	0.03	0.501	0.486	0.394
Liver (% SW)	1.83	1.84	1.85	1.88	2.02	2.04	1.93	2.04	0.01	0.01	0.01	0.375	0.551	0.216
Gut (% SW)	5.81	5.99	5.66	5.97	5.88	6.08	5.82	5.94	0.14	0.18	0.22	0.213	0.352	0.344
Gizzard (% SW)	2.06 ^b^	2.01 ^b^	1.93 ^b^	1.90 ^b^	3.42 ^a^	3.58 ^a^	3.34 ^a^	3.28 ^a^	0.11	0.29	0.34	0.158	0.024	0.011

Abbreviations: G, genotype; FS, farming system; G × FS, interaction between genotype and farming system; SW, slaughter weight; RTCC, ready-to-cook carcass; CC, cold carcass; SEM, standard error of the mean; BP, Bionda Piemontese; RM, Robusta Maculata; BP × S, Bionda Piemontese × Sasso; RM × S, Robusta Maculata × Sasso. Values with superscript ^a^, ^b^ are considered significantly different. *p*-values are considered significant when *p* < 0.05.

## Data Availability

The data presented in this study are available upon request from the corresponding author.

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
