# Peer review of "Productive Performances of Slow-Growing Chicken Breeds and Their Crosses with a Commercial Strain in Conventional and Free-Range Farming Systems"

_animals, 2023, doi:10.3390/ani13152540_

Round 1

Reviewer 1 Report

General comments

The topic of the manuscript is not new, but comparing different farming systems (conventional and organic) is interesting for the producers and consumers. In my opinion, the manuscript could be suitable for publication after revision, addressing the following comments:

 Introduction

Please formulate the research hypothesis.

 Keywords

Why intensive farming system has been omitted?

 Materials and methods

Line 127: Please remove redundant parentheses.

Line 133: Please remove a dot before [24].

Line 190: The Latin term should be italicized.

Line 257: Please be consistent in recording the P-value.

Please check the numbering of the tables. Table 2 cannot appear in the text before Table 1.

Please insert the spaces between values ​​and units.

 Results

Please check the numbering of the tables. Table 3 cannot appear in the text after the Figures 1 and 2.

Tables: Please explain how the statistical significance of differences was marked.

No reference to Table 4 in the text.

When the Authors write that mean values differ it should write level of significance (for example: "The value of sample A was higher (P<0.05) than that of sample B"). Please insert „P>0.05” when you write „No effect / no differences, etc.".

It seems that the analysis of the impact of the main factors (farming system and genotype), and then possible interactions, would be clearer.

Author Response

Dear reviewer,

Thank you very much for the precious inputs, I have tried to answer all the suggestions as to improve this work. I hope you find the manuscript satisfactory.

Introduction:

Added the research hypothesis, specifying that we expect the improve in growth performances thanks to crossbreeding but we want to investigate in particular these breeds in two different farming systems.

Materials and methods:

  • Line 127: removed redundant parentheses.
  • Line 133: removed dot before [24].
  • Line 190: Latin term is now italicized.
  • Corrected the orders of tables.

Results

  • Highlighted the statistical significance of differences.
  • Fixed the absence of table 4 in the text
  • Added the P>0.05” when written “No effect / no differences, etc.".

Reviewer 2 Report

The statistical analysis is not necessarily incorrect, but if you consider that they are repeated measures, why not estimate the most appropriate covariance structure for these data and make an analysis that contemplates these covariances between measures.

I believe that figures one and two would be easier to interpret through a line graph.

The conclusions are adequate, perhaps it is necessary to specify the implications that are put in this section.

Author Response

Dear reviewer,

Thank you very much for the precious suggestions. We chose the statistical approach of a GLM because it allowed us to take in consideration the replicate as a repeated measurements on the same pen. Also, after literature search, we saw that this is one of the mainly used approach and so we chose in order to be consistent with and comparable to previous studies. But we will definitely take into consideration your suggested approach.

We chose to express the data of figure 1 and 2 through histograms because the data of the two purebreds and crossbreds were very similar, consequently the lines of a line graph would overlap making them difficult to understand.

Reviewer 3 Report

Dear editor,

Thank you for giving me the possibility to review the paper entitled “Productive performances of slow growing chicken breeds and their crosses with a commercial strain in conventional and free-range farming systems” (animals-2492339).

The paper has some major concerns that still do not permit publication in its current state, so I suggest to reject it.

There is a basilar approach to the research and a not well written method, moreover there is not real novelty in it. I think that results are not well shown, and that an economic analysis linked to feed consumption and benefits should be added to this paper.

Below the comments

L127 – delete brackets

L130-147 I’m not sure that this part should be in the M&M section. Probably can be added to the introduction that must surely ne revised in order to better guide the reader to the aim of the paper.

Line 154-152 Not clear how authors proceeded. Hw many days? I suggest to revise it and, probably, a figure for the experimental design can help the author to be more clear and the reader to better understand it.

L163 – mean weight: which values?

L176-177 It is better to report temperature and humidity. They don’t steal space

L180-192 As reported before, it should be specified days, latitude, humidity.

L193-195 Report mortality

L211 reference not correct

Table 2 – Please report also energy value. You report standard and low imput but not the energy of the two rations.

L226 – 232 How did you calculate ADG and ADF?

231-232 – Is a repetition of what reported in L193-194

Figure 1 and 2 – Are there statistical differences? Not reported

Table 3 no measure units reported. There are statistical differences in SW? Please, add costs analysis to add something more to the paper.

L314-321 I think is not the aim of the paper

L321-323 It is a obvious concept, doesn’t give anything more to the discussion

There are some minor typo errors that should be corrected

Author Response

Dear reviewer,

Thank you for your interesting and stimulating suggestions. I have endeavoured to provide thorough responses to each of them. I would like to clarify that the objective of this paper is to provide precise information pertaining to these extensively studied breeds, while simultaneously addressing a significant research gap by examining their productive performance under two distinct breeding systems. We believe that the novelty of our study lies not in the practice of crossbreeding itself, but rather in the enhancement of productive performance in F1 outputs of these specific breeds. These animals are of great value and represent a vital resource for our geographical area, as evidenced by the numerous publications from other authors in the past.

Data presentation has been carried out following a linear model analysis, focusing on the three primary effects reported in the text and tables: genotype, farming system, and their interaction. This analytical and data presentation approach has been chosen due to its frequent utilization by many other authors, particularly in recent times.

Unfortunately, it was not possible to include an economic analysis, as it falls outside the expertise of any of the authors and was not considered during the project's writing phase. The primary goal was to provide data and examine animal behavior under different breeding systems, serving as a starting point for future research.

I tried to integrate all your suggestions and make the manuscript clearer where you thought it was not. I hope you find the manuscript improved. Thank you.

L 130-147 We decided to put this part in the material and methods section because these breeds are highly categorized and there’s previous literature on the subject and most of all they represent the starting point of our research.

L 154-152 The experimental design is divided in two phases: the first is the brood and it is described in the text The 1-d-chicks were moved to the experimental poultry facility and reared for the first 20 days in the brood which was divided into four pens (one for each genotype)”. I tried to make it clearer thanks to your advice specifying it lasted from hatching until day 20.

L 163 Added mean weights

L 176-177 As the temperature and relative humidity are commonly indicated by Aviagen for commercial broiler farming, we decided as many other authors previously did to just report the latest handbook that is available. The objective was to simulate as much as possible a controlled environment like the intensive farming systems as opposed to a natural free-range farming system. Also following Animals guidelines, we chose to not report the specific programs created by Aviagen but to simply report them as a bibliographic citation.

L 193-195 It is reported in the results in lines 261-263

L 211 Reference corrected

Table 2 – The value of metabolizable energy of each diet is reported in the table. I added the unit measure.

L 226 – 232 As reported the animals were weekly weighed, and feed consumption was registered and calculated based on the amount of feed consumed by the animals as standard.

L 231-232 Removed the repetition.

Figure 1 and 2 – Added the statistical differences indicating them with superscript letters.

Table 3 added the unit measure as requested. The statistical differences are indicated in the P-value section of the table and with superscript letters.

L 314-321 we actually wanted to give more updated information on growth performance of these local breeds as they are highly characterized, but no one previously investigated these breeds in organic/free range farming with a faster slaughtering time.

Round 2

Reviewer 3 Report

Dear Editor,

the suthors replied to all comments and improved their paper. it can be accepted for publication